# Relationship between Serum Angiopoietin-like Proteins 3 and 8 and Atherogenic Lipid Biomarkers in Non-Diabetic Adults Depends on Gender and Obesity

**DOI:** 10.3390/nu13124339

**Published:** 2021-11-30

**Authors:** Karolina Murawska, Magdalena Krintus, Magdalena Kuligowska-Prusinska, Lukasz Szternel, Anna Stefanska, Grazyna Sypniewska

**Affiliations:** Department of Laboratory Medicine, Collegium Medicum, Bydgoszcz, Nicolaus Copernicus University, 87-110 Torun, Poland; murawska.karolina05@gmail.com (K.M.); magda77@cm.umk.pl (M.K.-P.); l.szternel@cm.umk.pl (L.S.); zuzanna@cm.umk.pl (A.S.); odes@cm.umk.pl (G.S.)

**Keywords:** hypertriglyceridemia, cardiovascular risk, angiopoietin-like proteins

## Abstract

Hypertriglyceridemia is an independent risk factor for coronary artery disease. Lipoprotein lipase (LPL) plays an essential role in the metabolism of triglyceride-rich lipoproteins (TRLs). Angiopoietin-like proteins ANGPTL3 and ANGPTL8 are shown to be important regulators of LPL activity. Increased concentrations of these proteins may reflect cardiovascular risk, and the treatment of patients with dyslipidemia with ANGPTLs inhibitors may decrease this risk. We assessed the gender-specific relationships of serum ANGPTL3 and ANGPTL8 with atherogenic lipid biomarkers and obesity in non-diabetic adults. The study comprised 238 participants aged 25–74 [122 with triglycerides (TG) <150 mg/dL (<1.7 mmol/L) and 116 with hypertriglyceridemia]. Total cholesterol, HDL-cholesterol, LDL-cholesterol, TG, C-reactive protein (CRP), glycated hemoglobin, apolipoprotein B, small dense LDL-C (sd-LDL-C), ANGPTL3, and ANGPTL8 were measured. Non-HDL-cholesterol, remnant cholesterol (remnant-C) concentrations, and body mass index (BMI) were calculated. Results: Women and men did not differ in terms of age, CRP levels, the percentage of obese subjects, and concentrations of atherogenic lipid biomarkers, except higher TG in males and higher ANGPTL3 concentrations in females. Positive correlations of both ANGPTLs with TG, remnant-C, and sdLDL-C levels were found in females. In males, only ANGPTL3 correlated positively with atherogenic biomarkers, but there were no correlations with ANGPTL8. Concentrations of ANGPTL3 were higher in obese men, whereas ANGPTL8 levels were higher in obese women. In women alone, ANGPTL8 showed very good discrimination power to identify subjects with hypertriglyceridemia (AUC = 0.83). Contrary to this, ANGPTL3 was a better discriminator of hypertriglyceridemia (AUC = 0.78) in male subjects. Regression models, adjusted for age, sex, and BMI showed a weak but significant effect of ANGPTL8 to increase the risk of hypertriglyceridemia. Conclusions: In females, ANGPTL8 is more strongly associated with TRLs metabolism, whereas in males, ANGPTL3 plays a more important role. We suggest sex differences be taken into consideration when applying new therapies with angiopoietin-like proteins inhibitors in the treatment of dyslipidemia.

## 1. Introduction

The most common cause of cardiovascular diseases (CVDs) is coronary atherosclerosis and risk factors for the occurrence of CVDs include dyslipidemia, hypertension, obesity, diabetes, smoking, and limited levels of physical activity. Non-modifiable risk factors, which can also lead to the disease include a family history of coronary artery disease, gender, and age. Increased level of LDL-cholesterol is an important risk factor for the development of atherosclerosis, but high levels of triglyceride-rich lipoproteins (TRLs) also contribute to atherosclerotic cardiovascular disease (ASCVD) [1]. TRLs contain cholesterol, which accumulates in the atherosclerotic plaque. Increased triglyceride (TG) levels may then reflect increased levels of cholesterol in TRLs. Elevation of TG in the circulation is mostly caused by diabetes and obesity, but genetic factors may also contribute. Lipoprotein lipase (LPL) plays an essential role in the metabolism of TRLs. Recently, angiopoietin-like proteins ANGPTL3, ANGPTL4, and ANGPTL8 have been identified as important regulators of LPL activity involved in lipoprotein metabolism depending on nutritional status [2]. ANGPTL3 was shown to regulate circulating TG levels by inhibiting LPL activity and participating in the metabolism of cholesterol [2,3]. ANGPTL3 activity is enhanced by ANGPTL8 which interacts with ANGPTL3 by forming a complex increasing the inhibitory effect on LPL [4]. It was reported that ANGPTL8 levels are positively associated with obesity, diabetes, and dyslipidemia [3,5]. Observational studies have shown that increased concentrations of these proteins may reflect cardiovascular risk and suggest that treatment of patients with dyslipidemia with ANGPTL3 inhibitors may decrease the CVD risk [6,7,8]. Pharmacological (with monoclonal antibodies against ANGPTL3) and genetic (with ASO-antisense oligonucleotides) suppression of ANGPTL3, which markedly reduces concentrations of circulating atherogenic lipoproteins and may improve insulin sensitivity, makes ANGPTL3 a new emerging target for lipid-lowering therapy [9]. There are very few published studies on the relationship of angiopoietin-like proteins with disturbances in lipid metabolism, stratified by gender, with data currently contradictory and inconclusive [5]. In our study, we aimed to assess the gender-specific relationships of serum ANGPTL3 and ANGPTL8 with biomarkers of atherogenic lipid profile and obesity in non-diabetic adults.

## 2. Materials and Methods

### 2.1. Study Subjects

The study was designed as a cross-sectional study based on a subsample of a cohort of 600 apparently healthy Caucasians, females and males, aged between 18 and 75 years. Self-declaration of good health based on a questionnaire was previously published elsewhere [10]. Recruitment occurred at workplaces in the Kuyavian-Pomeranian Region, Poland. Subjects were free from any heart disease, had no arterial hypertension or diabetes, underwent no cardiac intervention, and did not undergo cardiac or lipid-lowering therapy. Participants with infections and previously diagnosed chronic inflammatory disease were also excluded. All participants gave written consent to be included in the study and completed a questionnaire prior to blood collection. The study protocol was approved by the Local Bioethics Committee in accordance with the Helsinki declaration (KB 348/21.04.2015, annex KB 490/25.06.2019). In order to define the study group a criterion based on TG concentration was employed. Finally, 238 participants were randomly selected, of which 122 (48 females and 74 males, aged 25–70) had TG < 150 mg/dL (<1.7 mmol/L), while 116 (47 females and 69 males, aged 25–74) had hypertriglyceridemia (TG ≥ 150 mg/dL; ≥1.7 mmol/L).

Fasting venous blood samples were collected between 7.30 a.m. and 10.00 a.m. and serum was obtained. Anthropometric measurements were performed on the same day as blood samples were taken, and BMI (body mass index) was calculated.

### 2.2. Laboratory Measurements

Total cholesterol (TC), HDL-cholesterol (HDL-C), LDL-cholesterol (LDL-C), TG, C-reactive protein (CRP), and glycated hemoglobin (HbA1c) were measured in serum, while whole EDTA blood samples were used to determine HbA1c. Apolipoprotein B (apoB), small dense low-density lipoprotein cholesterol (sdLDL-C), and angiopoietin-like proteins 3 and 8 (ANGPTL3, ANGPTL8) were assayed in previously deep-frozen serum samples. HbA1c and lipid profile parameters (TC, HDL-C, LDL-C, and TG) were determined using an Abbott ARCHITECT ci8200 instrument (Abbott, Wiesbaden, Germany). ApoB, sdLDL-C, and CRP concentrations were measured on a Pentra 400 analyzer (Horiba ABX, Montpellier, France), and HbA1c was measured on a Bio-Rad VARIANT II turbo with high-performance liquid chromatography (Bio-Rad Laboratories, California, US). Reagents for the sdLDL-C assay were provided by Randox Laboratories. All measurements were performed at the Department of Laboratory Medicine, Nicolaus Copernicus University, Collegium Medicum in Bydgoszcz, Poland. Non-HDL-cholesterol (non-HDL-C) and remnant cholesterol (remnant-C) concentrations were calculated using the following formulas: non-HDL-C = TC–HDL-C; remnant-C = TC-LDL-C-HDL-C [11]. The following cut-off values were accepted as abnormal: TC ≥ 190 mg/dL (TC ≥ 5.0 mmol/L); LDL-C ≥ 115 mg/dL (LDL-C ≥ 3.0 mmol/L); HDL-C < 40 mg/dl (HDL-C < 1.0 mmol/L); apo B ≥ 1 g/L; TG ≥ 150 mg/dL (TG ≥ 1.7 mmol/L), non-HDL-C ≥ 145 mg/dl (non-HDL-C ≥ 3.7 mmol/L), remnant-C ≥ 3.0 mg/dL (remnant-C ≥ 0.078 mmol/L) [12]. HbA1c ≥ 39 mmol/mol was considered a cutoff for defining prediabetes [13].

ANGPTL3 and ANGPTL8 determinations were performed using BioVendor ELISA tests (BioVendor, Brno, Czech Republic). The limit of detection of ANGPTL3 was 1.08 ng/mL, and the concentration range was up to 400 ng/mL with intra-assay precision of 1.8–5.6%, and inter-assay precision of 7.3–10.5%. The detection limit for ANGPTL8 was 0.244 ng/mL with intra-assay precision of 5.4–9.4%, and inter-assay precision of 2.4–5.3%. Samples with absorbance exceeding that of the highest standard were remeasured at a higher dilution.

### 2.3. Statistical Analysis

Variables with non-Gaussian distribution were presented as medians and 25th and 75th percentiles, and those with a normal distribution using means and ± standard deviation (SD). The Shapiro–Wilk test was applied to test the normality. Data were compared using Student’s *t*-test for normally distributed variables and the Mann–Whitney U test for non-normally distributed variables. Categorical variables were compared using the chi-squared test. Rho Spearman correlation analysis, receiver operating characteristics, and logistic regression analysis were also performed. Variables were logarithmically transformed before their introduction in logistic regression analysis in order to improve their adherence to the normal distribution. A comparison of independent ROC curves was performed. A *p* < 0.05 was considered statistically significant for all calculations. Statistical analyses were carried out using SPSS v.20 (Statistical Package for the Social Sciences version 20, Armonk, NY, USA) and PQStat v.1.6.8 (PQStat Software, Poznan, Poland).

## 3. Results

The characteristics of study participants are presented dependent on gender (Table 1). Overall, females and males did not differ greatly in terms of age; however, among females, 48% were at ≥50 years of age, while among males, this was 36%. The levels of TC, LDL-C, apoB, non-HDL-C, remnant-C, HbA1c, and CRP were not significantly different. Mean BMI value, median TG, and sdLDL-C concentrations were significantly higher in men, while mean HDL-C concentrations were significantly lower (Table 1). The proportion of obesity (BMI ≥ 30 kg/m^2^ was similar among females and males, i.e., 22.3% and 21.7%, respectively. In addition, 27.4% of females and 25.5% of males had HbA1c levels, which could suggest the presence of prediabetes (*p* = 0.740). It is worth noting that mean or median values of all measured (TC, LDL-C, apoB) or calculated (non-HDL-C) lipid parameters, except for TG and remnant-C levels, were above the accepted cut-offs for normolipidemia. The median value of ANGPTL3 was significantly higher in females than in males, though that of ANGPTL8 was comparable.

Correlation analysis between angiopoietin-like proteins (ANGPTL3 and ANGPTL8) levels and atherogenic lipid biomarkers revealed interesting gender-specific differences (Table 2). In females, significant positive associations of both, ANGPTL3 and ANGPTL8, with TG, remnant-C, and sdLDL-C levels were found, while there was no relationship with HDL-C. Moreover, ANGPTL8 concentration correlated positively weakly though significantly with CRP and BMI.

On the contrary, in males, only ANGPTL3 levels correlated positively with all atherogenic lipid biomarkers, except for HDL-C, whereas no correlations were found for ANGPTL8. In addition, solely ANGPTL3 levels were weakly but significantly related to BMI and CRP. Notably, in both groups, we found a strong positive correlation between TG levels and remnant-C, reflecting cholesterol in TRLs (r = 0.755; *p* < 0.001 and r = 0.827 *p* < 0.001 in females and males, respectively). A moderately negative relationship between TG levels and HDL-C (r = −0.346; *p* = 0.001) was observed irrespectively of gender. Only in females, TG levels correlated positively with age (r = 0.297; *p* = 0.004). 

We did not find any correlation between ANGPTL3 and ANGPTL8 in the study group overall (r = 0.057; *p* = 0.385), nor similarly in either the female or male groups (r = 0.072; *p* = 0.486 and r = 0.106; *p* = 0.209, respectively). 

Table 3 presents concentrations of assessed biochemical markers in normal weight (BMI < 25 kg/m^2^) and obese (BMI ≥ 30 kg/m^2^) subjects, stratified by gender. Median concentrations of ANGPTL3 were significantly higher only in obese males, compared with normal-weight ones. On the other hand, ANGPTL8 was significantly higher in obese females, while in males, ANGPTL8 values were similar regardless of their BMI. Obese females and males were characterized by significantly higher CRP and significantly lower HDL-C concentrations. Moreover, significantly higher TG values and remnant-C were observed in obese men, whereas in obese women, a tendency toward elevated TG levels was found.

Table 4 presents the ROC analysis performed for ANGPTL3 and ANGPTL8 and selected atherogenic lipid indices to discriminate between subjects with hypertriglyceridemia and those without. In females, ANGPTL8 was found to be of very good diagnostic ability in identifying subjects with hypertriglyceridemia (AUC = 0.83), similar to that of remnant-C and sdLDL-C (AUC = 0.87), though the power of discrimination of ANGPTL3 was much lower (AUC = 0.71). In contrast, in males, ANGPTL3 was proved a good discriminator between subjects with hypertriglyceridemia and those without (AUC = 0.78), with remnant-C presenting excellent discrimination (AUC = 0.92). ANGPTL8 had a significantly lower AUC in males, compared with females (AUC = 0.54 vs. AUC = 0.83; *p* < 0.0001) and showed no diagnostic ability in males to identify subjects with hypertriglyceridemia. 

The optimal cutoff values for serum ANGPTL8 were similar in both females and males at 5 ng/mL and 3.4 ng/mL, respectively, whereas those for ANGPTL3 were much higher for females than for males at 251 ng/mL and 146 ng/mL, respectively.

In the course of further analysis, we developed logistic regression models for the prediction of hypertriglyceridemia in females and males (Table 5). The models in the multivariable analysis revealed that ANGPTL8 and remnant cholesterol were the best positive predictors of hypertriglyceridemia in women. ANGPTL8 increased the risk of hypertriglyceridemia by almost 33% and remnant cholesterol by 24%, and these effects were highly significant. In men, remnant cholesterol was the best positive predictor, whereas the effect of ANGPTL3 was negligible, and that of ANGPTL8 did not reach statistical significance.

## 4. Discussion

The role of elevated TRLs levels in ASCVD, as well as interest in factors regulating lipoprotein metabolism in the context of the treatment of hypertriglyceridemia, prompted studies of angiopoietin-like proteins. Recent studies revealed an association between ANGPTL3 and ANGPTL8 with lipid biomarkers in adults, young obese non-diabetic men, children, and adolescents with obesity and metabolic syndrome. However, very few of them stratified their data by gender [5,6,14,15,16]. We selected circulating ANGPTL3 and ANGPTL8 to assess gender-specific relationships with obesity and biomarkers of atherogenic lipid profile in non-diabetic adults aged between 25 and 74 years. Both ANGPTL3 and ANGPTL8 interact as regulators of LPL activity, and both are targets for pharmacotherapeutic agents currently undergoing testing as potentially effective drugs in the treatment of hypertriglyceridemia. The most common lipid disorder in the general Caucasian population affecting >60% is hypercholesterolemia; however, the prevalence of hypertriglyceridemia is also quite high, amounting to some 27–30% and higher in men than in women [1,17,18]. In this study, the proportion of hypertriglyceridemia in females and males was representative of a Polish population. According to a recent study conducted in Poland to evaluate the prevalence of dyslipidemia in the adult population, elevated triglyceride levels (>150 mg/dL) were observed in 31% of men and 20% of women [17], while in our study, this proportion was almost the same, with hypertriglyceridemia present in 29.4% of men and 19.7% of women. We did not assess either thyroid function or insulin resistance in our study; however, referring to recent data from our group, we may assume that most of the subjects were euthyroid [19]. Moreover, in our study, both females and males had similar glycemic status according to HbA1c values. 

Our study found elevated levels of serum TC in 67.2% of subjects, decreased HDL-C in 16%, TG levels higher in men than in women, and almost 6% of study subjects having moderately elevated serum triglycerides (300–512 mg/dL; 3.39–5.78). The proportion of obese subjects among females and males was the same (22%), and both groups were similarly composed in terms of age, concentrations of atherogenic lipid biomarkers, glycemic status, and CRP levels. 

In our study group, overall weak but significant correlations were observed between both ANGPL3 and ANGPTL8 and TG concentrations (r = 0.329 and 0.287; *p* < 001, respectively). Following stratification by gender, significant positive correlations remained between ANGPTL3 and TG, apoB, and remnant-C in both groups; however, positive correlations between ANGPTL8 and atherogenic lipid biomarkers were demonstrated only in women. Our observation differed from previously published research which showed positive correlations between ANGPTL8 and lipid biomarkers TG, LDL-C, and negative with HDL-C in a population of almost 990 Japanese adults undergoing routine medical checkups [6]. It is worth noting that in this Japanese study, gender differences were not analyzed. 

Our study showed that concentrations of ANGPTL3 were significantly lower in males than in females and that obesity had a strong effect on increasing ANGPTL3 levels in males. This finding is in accord with a significant positive correlation between ANGPTL3 and BMI in men and concurs with other observations demonstrating elevated levels of ANGPTL3 in severely obese young non-diabetic men and severely obese adolescents [14,20]. Interestingly, we found ANGPTL8 concentrations to be significantly lower in obese adolescents [20]. 

In this study, obesity significantly increased ANGPTL8 concentrations only in females, and a positive correlation between ANGPTL8 and BMI in this group was observed. By contrast, in the Japanese study above, in which gender differences were not taken into consideration, obesity and dyslipidemia were positively related to circulating ANGPTL8 [6]. However, our study group was not directly comparable with the latter, as their average BMI of subjects was higher, and they presented with higher concentrations of TG and LDL-C and lower HDL-C levels. This may explain the sex-specific differences demonstrated by us in relation to the effect of obesity on ANGPTL8 levels. Indeed, earlier research assessing the relationship between ANGPTL8 and the risk of metabolic syndrome in adults indicated that ANGPTL8 was a good predictor of increased TG in females but not in males [5]. 

The diagnostic power of both these angiopoietin-like proteins to identify subjects with hypertriglyceridemia (TG ≥ 150 mg/dL; ≥1.7 mmol/L) was similarly dependent on gender. In females, ANGPTL8 showed a very good power of discrimination (AUC = 0.83), similar to that of remnant-C (AUC = 0.87), whereas in males, ANGPTL8 demonstrated no diagnostic ability (AUC = 0.54). On the contrary, in males, ANGPTL3 was a better discriminator between subjects with hypertriglyceridemia (AUC = 0.78) than in females (AUC = 0.71). This is in accord with the higher correlation coefficient of ANGPTL3 with TG in men. Among predictors of hypertriglyceridemia, ANGPTL8 was the best positive predictor in women, whereas the effect of ANGPTL3, though significant, was very low irrespective of gender. 

Sex differences in the relationships between ANGPTL3 and ANGPTL8 and atherogenic lipid biomarkers between women and men may be explained by the effect of estrogens in women. In premenopausal women, estrogens are associated with the reduced prevalence of arterial hypertension, coronary artery disease, myocardial infarction, and stroke. Differences in lipoprotein metabolism between women and men involve hormonal and genetics-mediated effects. The mechanisms are complex but it is well known that estrogens are involved in pathways governing lipoprotein metabolism in women, and it has also been demonstrated that lower levels of triglycerides in women result from higher LPL activity [21,22]. Aging is a factor that leads to an increased risk of changes in the lipid metabolism of women, with an increased risk of atherosclerotic cardiovascular disease. It is worth noting that in this study, the only positive correlation of both ANGPTL3 and TG with participants’ age was observed in women (r= 0.299; *p*= 0.003 and r = 0.297 *p* = 0.004, respectively). 

Sex hormones may also be involved in the mechanisms responsible for weight gain and changes in the accumulation of visceral fat and visceral obesity, which are associated with increased cardiometabolic risk. The differing influence of obesity in women and men on ANGPTL3 and ANGPTL8 concentrations could be explained by differences in body fat distribution as suggested in earlier studies [21,23]. This, however, is an assumption that cannot be borne out by the results of our study, as data on waist circumference or waist-to-hip ratio, helpful in the identification of obese subjects at high cardiometabolic risk, were not collected, and accordingly, this is a limitation of this study. 

## 5. Conclusions

Our study was observational in nature and included a relatively limited number of subjects. However, we were able to demonstrate gender-dependent differences in associations between ANGPTL3 and ANGPTL8 and atherogenic lipid biomarkers in non-diabetic adults over a wide age range. We suggest that in females, ANGPTL8 is more strongly associated with TG-rich lipoprotein metabolism than ANGPTL3, whereas in males, ANGPTL3 plays a more important role. Our results contribute to the understanding of gender differences in lipoprotein metabolism and potential differences in the efficacy of treating dyslipidemia with new pharmacological agents, employing inhibitors of angiopoietin-like proteins [24,25]. As there are currently several new therapeutical approaches for the treatment of hypertriglyceridemia being assessed, we advocate that sex differences should be taken into consideration when applying new therapies with angiopoietin-like proteins inhibitors in the treatment of dyslipidemia.

## Figures and Tables

**Table 1 nutrients-13-04339-t001:** General characteristics of study participants by gender.

Variables	Females *n* = 95	Males *n* = 143	*p*
Age (years)	47.0 ± 11.0	44.0 ± 11.0	0.052
BMI (kg/m^2^)	26.0 ± 4.0	27.0 ± 4.0	0.013
HbA1c (mmol/mol)	37.1 ± 4.5	36.7 ± 4.7	0.339
TC (mmol/L)	5.70 ± 1.19	5.40 ± 0.93	0.075
HDL-C (mmol/L)	1.50 ± 0.31	1.27 ± 0.28	<0.001
LDL-C (mmol/L)	3.47 ± 1.03	3.42 ± 0.88	0.985
TG (mmol/L)	1.63 (0.82–2.19)	1.65 (1.07–2.35)	0.047
ApoB (g/L)	1.03 ± 0.29	1.05 ± 0.23	0.364
CRP (mg/L)	0.94 (0.27–2.90)	0.61 (0.27–1.61)	0.067
sdLDL-C (mmol/L)	0.77 (0.55–1.33)	1.01 (0.73–1.41)	0.033
Non-HDL-C (mmol/L)	4.19 ± 1.22	4.14 ± 0.93	0.918
Remnant-C (mmol/L)	0.60 (0.41–0.85)	0.67 (0.49–0.93)	0.105
ANGPTL3 (ng/mL)	242 (199.8–300.7)	193.2 (100.7–250.6)	<0.001
ANGPTL8 (ng/mL)	5.8 (3.74–9.16)	6.8 (4.14–9.83)	0.295

Notes: means ± SD or medians (25th and 75th percentiles).

**Table 2 nutrients-13-04339-t002:** Spearman rank correlation coefficients between ANGPTL3 and ANGPTL8, and atherogenic biomarkers by gender.

Variables	ANGPTL3	ANGPTL8	ANGPTL3	ANGPTL8
	Females	Females	Males	Males
	*n* = 95	*n* = 95	*n* = 143	*n* = 143
TG	r = 0.378	r = 0.524	r = 0.471	r = 0.091
	*p* < 0.001	*p* < 0.001	*p* < 0.001	*p* = 0.282
HDL-C	r = −0.101	r = −0.121	r = −0.145	r = 0.005
	*p* = 0.328	*p* = 0.243	*p* = 0.085	*p* = 0.953
apoB	r = 0.304	r = 0.167	r = 0.185	r = −0.076
	*p* = 0.003	*p* = 0.106	*p* = 0.027	*p* = 0.367
sdLDL-C	r = 0.261	r = 0.389	r = 0.259	r = 0.060
	*p* = 0.011	*p* < 0.001	*p* = 0.002	*p* = 0.475
Non-HDL-C	r = 0.323	r = 0.198	r = 0.194	r = −0.123
	*p* < 0.001	*p* = 0.054	*p* = 0.021	*p* = 0.142
Remnant-C	r = 0.259	r = 0.374	r = 0.338	r = 0,038
	*p* = 0.011	*p* < 0.001	*p* = 0.001	*p* = 0.660
CRP	r = −0.159	r = 0.315	r = 0.196	r = −0.054
	*p* = 0.124	*p* = 0.002	*p* = 0.019	*p* = 0.521
BMI	r = −0.046	r = 0.248	r = 0.238	r = 0.112
	*p* = 0.659	*p* = 0.015	*p* = 0.004	*p* = 0.185
Age	r = 0.299	r = 0.100	r = −0.042	r = −0.018
	*p* = 0.003	*p* = 0.333	*p* = 0.616	r = 0.828

Notes: r—correlation coefficient; *p*—statistical significance.

**Table 3 nutrients-13-04339-t003:** Concentrations of assessed biochemical markers in normal weight (BMI < 25 kg/m^2^) and obese (BMI ≥ 30 kg/m^2^) subjects, stratified by gender.

Variables	Females BMI < 25 kg/m^2^ (*n* = 43)	Females BMI ≥ 30 kg/m^2^ (*n* = 21)	*p*	Males BMI < 25 kg/m^2^ (*n* = 37)	Males BMI ≥ 30 kg/m^2^ (*n* = 30)	*p*
Age (years)	45 (37–55)	47 (42–54)	0.645	40 (31–52)	42 (34–53)	0.514
HbA1c (mmol/mol Hb)	37.7 (34.9–39.3)	38.8 (35.5–39.9)	0.321	35.5 (32–38.8)	37.7 (33.4–41)	0.057
CRP (mg/L)	0.6 (0.2–2.1)	2.5 (1.2–3.5)	<0.001	0.4 (0.1–0.8)	1.2 (0.5–2.6)	<0.001
TC (mmol/L)	5.59 (4.69–6.92)	5.28 (4.84–5.96)	0.178	5.18 (4.58–5.83)	5.0 (4.59–5.67)	0.581
LDL-C (mmol/L)	3.42 (2.67–4.58)	3.11 (2.54–3.68)	0.261	3.13 (2.74–3.86)	3.24 (2.59–3.81)	0.925
HDL-C (mmol/L)	1.53 (1.35–1.84)	1.30 (1.19–1.55)	0.008	1.35 (1.19–1.61)	1.14 (1.01–1.22)	0.001
TG (mmol/L)	1.19 (0.79–2.05)	1.82 (1.19–2.28)	0.081	1.20 (0.94–1.45)	1.97 (1.54–2.86)	0.001
apoB (g/L)	1.06 (0.84–1.23)	0.94 (0.81–1.10)	0.352	0.94 (0.82–1.14)	1.03 (0.90–1.13)	0.274
Remnant-C (mmol/L)	0.52 (0.39–0.78)	0.65 (0.54–0.85)	0.226	0.52 (0.39–0.75)	0.78 (0.54–0.93)	0.023
sdLDLC (mmol/L)	0.70 (0.52–1.29)	0.98 (0.28–1.32)	0.435	0.85 (0.62–1.04)	1.09 (0.85–1.45)	0.013
non-HDL-C (mmol/L)	4.17 (3.16–5.52)	4.07 (3.24–4.51)	0.491	3.63 (3.19–4.43)	3.89 (3.47–4.56)	0.426
ANGPTL3 (ng/mL)	246 (214–303)	238 (194–301)	0.518	118 (64–212)	207 (140–257)	0.010
ANGPTL8 (ng/mL)	4.7 (3.4–6.4)	6.6 (4.8–8.8)	0.031	6.4 (3.6–10.3)	7.4 (4.7–9.9)	0.343

Notes: medians (25th and 75th percentiles); *p*—statistical significance.

**Table 4 nutrients-13-04339-t004:** ROC analysis and diagnostic ability of evaluated biomarkers to predict hypertriglyceridemia, stratified by gender.

	Females			Males			
Biomarker	AUC (95% CI)	Cutoff	Sensitivity	AUC (95% CI)	Cutoff	Sensitivity	*p* F vs. M
ANGPTL8 ng/ml	0.83 (0.74–0.90)	>5	0.85	0.54 (0.45–0.62)	>3.4	0.91	0.001
ANGPTL3 ng/mL	0.71 (0.60–0.80)	>251	0.64	0.78 (0.71–0.85)	>146	0.94	0.271
sdLDL-C mmol/L	0.87 (0.79–0.93)	>0.73	0.85	0.84 (0.77–0.90)	>43	0.71	0.534
Remnant-C mmol/L	0.87 (0.78–0.93)	>0.60	0.81	0.92 (0.86–0.96)	>0.67	0.88	0.255
non-HDL-C mmol/L	0.77 (0.67–0.85)	>3.86	0.81	0.65 (0.56–0.73)	>3.44	0.90	0.067

Notes: area under the curve (AUC); confidence interval (CI); F—females; M—males; *p*—statistical significance.

**Table 5 nutrients-13-04339-t005:** Multivariable logistic regression analyses for predictors of hypertriglyceridemia stratified by gender.

	ORs (95% CI) Per One Unit Increase			
Variables	Females	*p*	Males	*p*
TC	1.024 (1.010–1.037)	0.001	1.010 (0.999–1.020)	0.055
sdLDL-C	1.080 (1.059–1.160)	<0.001	1.100 (1.078–1.137)	<0.001
Remnant-C	1.240 (1.120–1.360)	<0.001	1.285 (1.182–1.397)	<0.001
ANGPTL3	1.012 (1.005–1.020)	<0.001	1.006 (1.002–1.010)	0.002
ANGPTL8	1.327 (1.136–1.551)	<0.001	1.009 (0.998–1.086)	0.800

All models were adjusted for age and BMI: Females TC R2 Cox–Snell = 0.20; R2 Nagelkerke = 0.26; sdLDL-C R2 Cox–Snell = 0.40; R2 Nagelkerke = 0.54; remnant-C R2 Cox–Snell = 0.43; R2 Nagelkerke = 0.57; ANGPTL3 R2 Cox–Snell = 0.17; R2 Nagelkerke = 0.22; model ANGPTL8 R2 Cox–Snell = 0.24 R2 Nagelkerke = 0.32. Males TC R2 Cox–Snell = 0.15; R2 Nagelkerke = 0.20; sdLDL-C R2 Cox–Snell = 0.39; R2 Nagelkerke = 0.52; remnant-C R2 Cox–Snell = 0.54; R2 Nagelkerke = 0.71; ANGPTL3 R2 Cox–Snell = 0.19; R2 Nagelkerke = 0.26; model ANGPTL8 R2 Cox–Snell = 0.12 R2 Nagelkerke = 0.17. ORs—odd ratios; CI—confidence interval.

## Data Availability

The data presented in this study are available on request from the corresponding author.

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
