# Peer review of "Relationship between Serum Angiopoietin-like Proteins 3 and 8 and Atherogenic Lipid Biomarkers in Non-Diabetic Adults Depends on Gender and Obesity"

_nutrients, 2021, doi:10.3390/nu13124339_

Round 1

Reviewer 1 Report

In the manuscript “Relationship between serum angiopoietin-like proteins 3 and 8 and atherogenic lipid biomarkers in non-diabetic adults depends on gender and obesity”, Murawska et al. assessed the gender specific relationship between ANGPTL3 and 8 proteins and lipid biomarkers. The study is of potential interest in the field, particularly due to the limited understanding of gender differences in lipid metabolism. The results are clearly presented, and the manuscript is well-written. My major concern about the study is its sample size. The study included 95 women and 143 men, and the sample size is further reduced in the BMI stratified analyses. Other concerns related to the analyses are:

  1. More details on the statistical analyses are required for clarity. As many of the variables are not normally distributed, it is not clear if the authors transformed the data to reach the approximate normality, particularly for regression analysis.
  2. It is not clear if the logistic models used were multivariate or multivariable? What is the difference in the multivariate and univariate analysis presented in Table 5. Moreover, as most of the analysis in the manuscript are performed separately in men and women, it would be more appropriate to perform logistic regression analysis in Table 5 separately in men and women.
  3. ANGPTL8 interacts with ANGPTL3 increasing the inhibitory effect on LPL and levels of TG. It would be interesting to include interaction term ANGPTL8* ANGPTL3 to see if interaction in circulating ANGPTL8 and ANGPTL3 levels associate with TG levels or increased risk for hypertriglyceridemia.
  4. As many factors could affect the relationship between ANGPTL8 and ANGPTL3 and other lipid biomarkers such as age and BMI, it would be more appropriate to perform the regression analysis adjusting for suitable covariates.
  5. Authors should provide exact p values for all the tests.

Author Response

We thank very much for all valuable  reviewers’ comments and suggestions.

We agree on the relatively limited sample size however, our study group was selected from the healthy cohort stringently selected based on the predefined criteria such as  HbA1c <48 mmol/mol, BNP <35 ng/L, and eGFR >60 mL/min/1.73m2. Finally the study group comprised of non-diabetic adults over a wide age range (females 25-70 years; males 25-74 years) presenting with a proportion of  hypertriglyceridemia that was representative for population in Poland. According to the recent study conducted in Poland to evaluate the prevalence of dyslipidemia in the population over 18 years of age elevated triglyceride levels (>150 mg/dL) were observed in 31% of men and 20% of women (ref. 16) whilst in our study this proportion was almost the same : hypertriglyceridemia was present in 29.4% of men  and in 19.7% of women.

We are aware of this limitation and we  addressed „the relatively limited sample size”  in the Conclusions

Moreover, in the revised version we included a comment on this in the Discussion „In the present study the proportion of  hypertriglyceridemia in females and males was representative of a Polish population. According to a recent study conducted in Poland to evaluate the prevalence of dyslipidemia in the adult population elevated triglyceride levels (>150 mg/dL) were observed in 31% of men and 20% of women [16], whilst in our study this proportion was almost the same with hypertriglyceridemia present in 29.4% of men and 19.7% of women”.

  1. More details on the statistical analyses are required for clarity. As many of the variables are not normally distributed, it is not clear if the authors transformed the data to reach the approximate normality, particularly for regression analysis.

Thank you for this important comment. We have added this to the Statistical analysis:

Variables were logarithmically transformed before their introduction in logistic regression analysis in order to improve their adherence to the normal distribution.

  1. It is not clear if the logistic models used were multivariate or multivariable? What is the difference in the multivariate and univariate analysis presented in Table 5. Moreover, as most of the analysis in the manuscript are performed separately in men and women, it would be more appropriate to perform logistic regression analysis in Table 5 separately in men and women.

We thank very much for this comment although „the terms 'multivariate analysis' and 'multivariable analysis' are often used interchangeably in medical and health sciences research. However, for sure our regression analysis was multivariable. Multivariable regression models are used to establish the relationship between a dependent variable (i.e. an outcome of interest, which is hypertriglyceridemia) and more than 1 independent variable.  In accordance with the reviewers’ suggestion in the revised version we include a new multivariable regression analysis performed after stratification by gender. As a consequence we have removed the previous Table 5 from the original version of the manuscript and in the revised version included a new Table 5 with multivariable regression analyses that were adjusted for BMI and age.

  1. ANGPTL8 interacts with ANGPTL3 increasing the inhibitory effect on LPL and levels of TG. It would be interesting to include interaction term ANGPTL8* ANGPTL3 to see if interaction in circulating ANGPTL8 and ANGPTL3 levels associate with TG levels or increased risk for hypertriglyceridemia.

Morinaga et al. in their large sample study have found very weak, though significant correlation,  between ANGPTL3 and ANGPTL8 (-0.105; p=0.001). In the article  of Morinaga et al. (ref.6) they reported on the association between TG and log ANGPTL3*logANGPTL8 in the linear model and they found β was much lower, though significant,  for this interaction than for ANGPTL8 alone . Moreover, in this large sample study (n=988) that was not stratified by gender the association between TG and log ANGPTL3 was very low and insignificant (β =0.003; p=0.943).

We did not find any correlation between both ANGPTLs in the study group overall nor similarly in either the female or male groups. Therefore, we did not perform a linear regression analysis aimed to evaluate interactions between both ANGPTLs. In the revised version, we included the data on correlations between both ANGPTLs.

  1. As many factors could affect the relationship between ANGPTL8 and ANGPTL3 and other lipid biomarkers such as age and BMI, it would be more appropriate to perform the regression analysis adjusting for suitable covariates.

We agree with the reviewers’ suggestion and in the new Table 5 – multivariable analysis were adjusted for age and BMI.

  1. Authors should provide exact p values for all the tests.

We provided the exact p values for all tests, however we rounded it to 3 decimals. Consequently, p values smaller than 0.001 we reported as p<0.001.

Reviewer 2 Report

Summary: This study showed the sex differences of the ANGPTL8 and ANGPTL3 levels. Concentrations of ANGPTL3 were higher in obese men, whereas ANGPTL8 levels were higher in obese women. In women alone, ANGPTL8 showed excellent discrimination power to identify subjects with hypertriglyceridemia (AUC=0.83). Contrary to this, ANGPTL3 was a better discriminator of hypertriglyceridemia (AUC=0.78) in male subjects. Regression models, adjusted for age, sex and BMI showed a weak but significant effect of ANGPTL8 to increase the risk of hypertriglyceridemia.

Although this is a study with not many patients, it is well designed and well written. The conclusion is appropriate, and the methods are clear and well done. Most importantly, this sex difference of ANGPTL8 and ANGPTL3 has important clinical implications since medications against these two proteins are being developed.

Author Response

We are grateful to the reviewer for his positive comments on our study.

Reviewer 3 Report

The authors aimed in this manuscript to assess the gender-specific relationships of ANGPTL3 and ANGPTL8 in with biomarkers of atherogenic lipid profile and obesity in non-diabetic adults. 

The paper is generally well written and results are promising. However some few points may  be further clarified:

1) Due to the well-known relationship between circulating Angptl3 and Angptl8 levels with Hypothyroidism, I suggest to clarify how many patients in the both groups had hypothyroidism in the both groups.

2) Basing on the well-known relationship between Insulin resistance and ANGPTLs ; I ask if an assessment of insulin resistance (using HOMA-Index ) in these patients was performed? Any differences between the 2 groups? This is an important point since insulin resistance in absence of Diabetes is a common finding in obesity.

Author Response

The authors aimed in this manuscript to assess the gender-specific relationships of ANGPTL3 and ANGPTL8 in with biomarkers of atherogenic lipid profile and obesity in non-diabetic adults.

The paper is generally well written and results are promising. However some few points may  be further clarified:

  • Due to the well-known relationship between circulating Angptl3 and Angptl8 levels with Hypothyroidism, I suggest to clarify how many patients in the both groups had hypothyroidism in the both groups.

Thank you very much for your comments. We would like to clarify the point related to hypothyroidism.

Various studies have reported the incidence of hypothyroidism to be up to 5% and   subclinical hypothyroidism to be between 3-15% depending on the population studied, with much higher frequency in women than in  men, particularly after 60 years of age.

Unfortunately we did not perform thyroid function tests in our study group – the study was based on a subsample of apparently healthy (self-declaration of good health) women and men, recruited on voluntary basis  at workplaces. Our study group was selected from the healthy cohort stringently selected based on the criteria such as  HbA1c <48 mmol/mol, BNP <35 ng/L, and eGFR >60 mL/min/1.73m2. All participants completed a detailed questionnaire prior to blood collection.

 Recent data from Poland shows that hypothyroidism is present in approx. 5% of females and 1% of males and Hashimoto thyroiditis occurs in 1.4-8% of women before the age of  60 years and in 7-17% of women after 60 years of age.

Our study group included women of which 11.6% were over 60 years of age therefore taking into consideration the highest prevalence of Hashimoto disease, in our study group only 2 females >60 years of age  and 6 females <60 years of age could possibly present with this disorder. This is however only a speculation and to support the  explanations we would like to present other recent data from our group that involved over 180 women with normoglycemia and 165 women with dysglycemia recruited from the same region in which TSH, blood lipids, HbA1c and CRP were assayed and HOMA-IR as well as the occurrence of  obesity were assessed (Kubacka J, Cembrowska P, Sypniewska G, Stefanska A. The Association between Branched-Chain Amino Acids (BCAAs) and Cardiometabolic Risk Factors in Middle-Aged Caucasian Women Stratified According to Glycemic Status. Nutrients. 2021 Sep 22;13(10):3307). In both studies the groups of  women were of  similar mean age, similar average lipid values, HbA1c and  CRP. In the recent study  similar median TSH concentration of 1.4(1.1-2.0)mIU/L in the normoglycemic females and 1,4 (1.0-2.1) mIU/L in the dysglycemic ones were found. These median values  were  below the 95th percentile for general population and very close to the average TSH concentration in healthy population reported as 1.5 mIU/L.

Therefore, with high certainty we may assume that women in our study group were almost all euthyroid. As the prevalence of hypothyroidism in men in Polish population  is approx. 7-fold lower we may also assume that  apparently healthy men in our study were euthyroid.

  • Basing on the well-known relationship between Insulin resistance and ANGPTLs ; I ask if an assessment of insulin resistance (using HOMA-Index ) in these patients was performed? Any differences between the 2 groups? This is an important point since insulin resistance in absence of Diabetes is a common finding in obesity.

Thank you very much for this important  comment and question.

We totally agree that insulin resistance is an important issue when assessing the effects of obesity on glucose and lipid metabolism. Unfortunately, we did not assess insulin resistance (HOMA-IR) in this study as we focused mainly on the assessment of the relationship between ANGPTLs and biomarkers of atherogenic lipid profile.  We measured  only HbA1c to exclude diabetes  and we did  not assess other parameters of  glucose homeostasis. In our study group of apparently healthy subjects  27.4% of females and 25.5% of males had  HbA1c level which could suggest the presence of prediabetes (American Diabetes Association. 2. Classification and Diagnosis of Diabetes: Standards of Medical Care in Diabetes-2018. Diabetes Care. 2018 Jan;41(Suppl 1):S13-S27. doi: 10.2337/dc18-S002).  These results may suggest that both men and women had similar glycemic status. For explanation of insulin-resistance state  we may only refer to the abovementioned previous study (Kubacka et al.)  from our group in which HOMA-IR was calculated. In both studies women were recruited from the same region of the country and women were of  very similar  mean age, moreover in the previous study among women with normoglycemia (n=180)  the percentage of obese subjects (BMI>30 kg/m2) was even higher (30%) than in the present  study in which obesity was found in 22% of  women. In the referenced study median HOMA-IR in normoglycemic females  was 1.11(0.79-1.56) that was lower than the accepted cut-off  for insulin resistant state  of >2.3 (Radikova, Z.; Koska, J.; Huckova, M.; Ksinantova, L.; Imrich, R.; Vigas, M.; Trnovec, T.; Langer, P.; Sebokova, E.; Klimes, I. Insulin sensitivity indices: A proposal of cut-off points for simple identification of insulin-resistant subjects. Exp. Clin. Endocrinol. Diabetes 2006, 114, 249–256). We cannot speculate on the frequency of insulin resistance in the present study as the data we have for comparison comes only  from the group of women.

We have added prediabetes criteria to the Material and Methods section.

In the Results we added a sentence: 27.4% of females and 25.5% of males had HbA1c levels which could suggest the presence of prediabetes (p=0.740).

In the Discussion we added a statement referring to both comments of the reviewer.

We did not assess neither thyroid function nor insulin resistance in our study, however, referring to recent data from our group we may assume that most of subjects were euthyroid (Kubacka J, Cembrowska P, Sypniewska G, Stefanska A. The Association between Branched-Chain Amino Acids (BCAAs) and Cardiometabolic Risk Factors in Middle-Aged Caucasian Women Stratified According to Glycemic Status. Nutrients. 2021 Sep 22;13(10):3307).

 We do hope our explanations are sufficient and fulfill reviewers’ expectations.